# A Network of 17 Microtubule-Related Genes Highlights Functional Deregulations in Breast Cancer

**DOI:** 10.3390/cancers15194870

**Published:** 2023-10-06

**Authors:** Sylvie Rodrigues-Ferreira, Morgane Morin, Gwenn Guichaoua, Hadia Moindjie, Maria M. Haykal, Olivier Collier, Véronique Stoven, Clara Nahmias

**Affiliations:** 1Gustave Roussy Cancer Center, F-94800 Villejuif, France; sylvie.rodrigues-ferreira@gustaveroussy.fr (S.R.-F.); morgane.morin@gustaveroussy.fr (M.M.); hadia.moindjie4@uvsq.fr (H.M.); maria.haykal@gustaveroussy.fr (M.M.H.); 2INSERM U981, Université Paris-Saclay, F-94800 Villejuif, France; 3Inovarion, F-75005 Paris, France; 4CBIO (Centre de Bioinformatique), Mines Paris-PSL, PSL Research University, F-75005 Paris, France; gwenn.guichaoua@minesparis.psl.eu; 5INSERM U900, Institut Curie, F-75005 Paris, France; 6MODAL’X, UPL, Université Paris Nanterre, CNRS, F-92000 Nanterre, France; ocollier@parisnanterre.fr

**Keywords:** Aurora kinases, biomarker, kinesins, mitotic defects, prognostic value, Systems Biology, therapeutic targets

## Abstract

**Simple Summary:**

The microtubule cytoskeleton is a key component of the cell and an important target for breast cancer therapy. Microtubule organization and function are tightly regulated by a panel of microtubule-related proteins (MT-Rel) to ensure cellular homeostasis. Deregulation of MT-Rel genes is likely to impact microtubule dynamics and subsequent cell functions. In this study, we evaluate the prognostic value of a panel of 17 MT-Rel genes in breast tumors and the functional consequence of their deregulation using a Systems Biology approach. This study highlights MT-Rel as potential prognostic biomarkers and interesting therapeutical targets to evaluate in breast cancer.

**Abstract:**

A wide panel of microtubule-associated proteins and kinases is involved in coordinated regulation of the microtubule cytoskeleton and may thus represent valuable molecular markers contributing to major cellular pathways deregulated in cancer. We previously identified a panel of 17 microtubule-related (MT-Rel) genes that are differentially expressed in breast tumors showing resistance to taxane-based chemotherapy. In the present study, we evaluated the expression, prognostic value and functional impact of these genes in breast cancer. We show that 14 MT-Rel genes (*KIF4A*, *ASPM*, *KIF20A*, *KIF14*, *TPX2*, *KIF18B*, *KIFC1*, *AURKB*, *KIF2C*, *GTSE1*, *KIF15*, *KIF11*, *RACGAP1*, *STMN1*) are up-regulated in breast tumors compared with adjacent normal tissue. Six of them (*KIF4A*, *ASPM*, *KIF20A*, *KIF14*, *TPX2*, *KIF18B*) are overexpressed by more than 10-fold in tumor samples and four of them (*KIF11*, *AURKB*, *TPX2* and *KIFC1*) are essential for cell survival. Overexpression of all 14 genes, and underexpression of 3 other MT-Rel genes (*MAST4*, *MAPT* and *MTUS1*) are associated with poor breast cancer patient survival. A Systems Biology approach highlighted three major functional networks connecting the 17 MT-Rel genes and their partners, which are centered on spindle assembly, chromosome segregation and cytokinesis. Our studies identified mitotic Aurora kinases and their substrates as major targets for therapeutic approaches against breast cancer.

## 1. Introduction

Breast cancer is a leading cause of death by malignancy in women worldwide. The classification of breast tumors into distinct histological and molecular subtypes has been paramount in orienting clinicians in their decision to deliver appropriate treatments to the patients. While hormonotherapy and HER2-targeted therapy are considered as treatments of choice for patients with luminal—estrogen receptor (ER)-positive—and HER2-amplified breast cancer subtypes [1], few therapies are available for breast tumors that become resistant to treatment or for triple-negative breast cancer (TNBC), that do not express ER nor HER2 receptors. In addition to chemotherapy, immunotherapy and targeted therapies are being developed for TNBC [2], but additional therapeutic strategies are warranted for this aggressive type of cancer. In the rapidly developing area of precision medicine, with the objective to deliver the right treatment to the right patient, it is of utmost importance to identify new prognostic and predictive biomarkers in order to select patients who need close medical follow up and adapted treatments [3].

For more than 50 years, attention has been paid to the microtubule cytoskeleton in the fight against cancer. Mitotic poisons—including taxanes, which bind and stabilize microtubules—have been used in combination with DNA targeting agents for breast cancer chemotherapy. Microtubules are essential components of the cytoskeleton involved in different steps of cell division, migration and intracellular transport of proteins and organelles, all these processes being deregulated in cancer. Microtubules are polarized structures formed by the assembly of tubulin dimers that rapidly alternate between phases of polymerization and depolymerization at the microtubule ends, in a process called dynamic instability [4,5].

The dynamic property of microtubules is crucial to their function. It ensures proper assembly of the mitotic spindle during mitosis and correct attachment of chromosomes to the kinetochores in metaphase to promote equal chromosome segregation during cell division. Microtubule dynamicity is essential for intracellular transport and is also required to establish cell polarity, explore the cytosol and target the cell cortex during migration. Microtubule assembly, dynamics and functions are tightly regulated by a large panel of microtubule-associated and regulatory proteins including MAPs, kinesins and kinases [6,7,8]. Defects in the expression or function of these microtubule-related proteins (designated MT-Rel) may thus provoke major cellular alterations with subsequent consequences on cancer initiation or progression [9,10,11]. Resistance to chemotherapeutic agents targeting the microtubule cytoskeleton may also result from dysregulation of MT-Rel, among other molecular mechanisms [9,12].

In a recent study, we took advantage of large databases containing both molecular and clinical data for breast cancer patients to investigate whether MT-Rel may represent new predictive biomarkers of breast cancer chemoresistance [13]. To this end, we compared the expression levels of a panel of 280 MT-Rel encoding genes in transcriptomic studies from three independent cohorts of patients with breast tumors classified as sensitive or resistant to chemotherapy. Bioinformatics studies identified a total of 117 MT-Rel genes that were significantly deregulated in sensitive versus resistant breast tumors, among which 17 genes were deregulated in all three cohorts of patients [13].

In the present study, we examined the possibility that these 17 predictive MT-Rel genes may be connected and represent a functional network for breast cancer prognosis. To this end, we examined expression levels of these genes and their co-regulation in mammary tumors and adjacent normal tissues. We also evaluated their potential value as prognostic biomarkers of patient survival and the functional consequence of their depletion on cell viability. Finally, we used a Systems Biology method to highlight gene networks potentially associated with important biological functions altered in breast cancer.

## 2. Materials and Methods

### 2.1. Gene Expression and Kaplan-Meier Analyses

MT-Rel mRNA expression levels were analyzed using the TNMplot database (http://www.tnmplot.com) [14] of 112 breast tumor samples and their corresponding adjacent normal tissues. Comparison of normal and tumor samples was performed by the Mann–Whitney U test.

Expression values of each MT-Rel gene were downloaded from TNMplot for correlation analysis. Unsupervised hierarchical clustering was performed using the JMP v7.0.1 software. Pearson correlation coefficients were calculated using GraphPad Prism9 software.

Association of MT-Rel expression with Overall Survival (OS) and Relapse-Free Survival (RFS) was determined using Kaplan–Meier plotter database (http://www.kmplot.com) [15]. For each gene, the best probeset was used (Jetset) and the best cut-off was selected to distinguish between tumors expressing low and high levels of the gene.

### 2.2. RNAi Screen and Cancer Dependency Map

MDA-MB-231 and MDA-MB-468 breast cancer cell lines were cultured as described [16,17]. Cells were transiently transfected with siRNA library (Dharmacon ON-TARGETplus) targeting each of the 14 overexpressed MT-Rel genes (4 individual siRNAs per gene) and a non-targeting control pool siRNA. Before primary screening, pilot experiments were performed in which siRNA doses and incubation times were titrated to optimize assay responses to negative and positive controls. Transfection was conducted in 96-well plates (5000 cells/well) containing 0.2 pmol of siRNA/well using Lipofectamine® RNAiMAX transfection reagent (Invitrogen, Life Technologies, Villebon sur Yvette, France). Cell viability was determined after 96 h using CellTiter-Glo luminescent cell viability assay (Promega, Charbonnières-les-bains, France).

The dependency data used in this manuscript were derived from the publicly available dataset (DepMap Public 23Q2+Score, Chronos), consisting of dependency data for our 14 genes of interest across 47 breast cancer cell lines. These data are available online at https://depmap.org/portal/download/custom/.

### 2.3. Systems Biology

The goal was to build a small and interpretable network connecting the 17 differentially expressed MT-Rel genes in order to highlight functional interplays and cellular processes that are deregulated in breast cancer. We used the Python module pypath 0.14.48 to process post-translational activity flow databases where we searched for direct interactors known for each of the 17 genes. This did not allow connecting the 17 genes into a single network, leaving *MTUS1* and *ASPM* unconnected. To address this issue, we “manually” connected MTUS1 based on known interactions in available scientific publications. Indeed, *MTUS1* gene product ATIP3 was shown to interact with KIF2A [18]. ASPM was connected thanks to its first neighbor CDK4, known to interact with CDK1. We observed that some of the network nodes that have been extensively studied, such as AURKB, MAPT, RACGAP1 and STMN1, had many first neighbors that were otherwise not connected to any other node of the network. In order to build the smallest possible network with only relevant information related to breast cancer, among these direct neighbors, we only retained the genes belonging to the 280 MT-related genes that are differentially expressed in tumor versus normal samples. This led to a network of 43 nodes and 87 edges involving the 17 genes. We completed this network by adding all interactions known between the 43 nodes of the network (41 genes, including 2 complexes). In some cases in which contrary information exists in the literature, we choose the interactions of the SIGNOR 3.0 database (https://signor.uniroma2.it) [19]. Network analysis unveiled several sub-networks corresponding to specific cellular processes, labeled based on Gene Ontology (GO) biological process enrichment analysis conducted using Gprofiler (https://biit.cs.ut.ee/gprofiler). Enrichment analysis was conducted on the 41 genes of the network with the Benjamini–Hochberg FDR correction.

## 3. Results

### 3.1. Regulation of Expression of 17 MT-Rel Genes in Breast Tumors 

A total of 17 genes encoding microtubule-related proteins (MT-Rel) were selected in a previous search for potential biomarkers of breast cancer chemoresistance [13]. To get further insight into the potential regulation and biological relevance of these 17 genes in breast cancer, we compared the expression levels of each gene in 112 tumors relative to adjacent normal tissues, using RNA-seq analyses available in public databases (www.tnmplot.com). As shown in Figure 1A and Appendix A, all genes were significantly differentially expressed in tumors compared to paired normal breast tissues. 

Fourteen genes (*KIF4A*, *ASPM*, *KIF20A*, *KIF14*, *TPX2*, *KIF18B*, *KIFC1*, *AURKB*, *KIF2C*, *GTSE1*, *KIF15*, *KIF11*, *RACGAP1*, *STMN1*) were up-regulated in tumors compared to normal tissues, with fold changes ranging from 2.5 to 17.8 (Figure 1B and Appendix A). Each of these 14 genes was overexpressed (expression level higher than 75% of normal levels) in more than 80% of breast tumors (Figure 1C and Appendix A). Six of them (*KIF4A*, *ASPM*, *KIF20A*, *KIF14*, *TPX2*, *KIF18B*) warrant particular attention as they show increased expression by more than 10-fold in tumor samples compared to normal tissues (Figure 1B and Appendix A). 

The *MAPT* gene was moderately but significantly up-regulated (fold change 1.44) in malignant compared to adjacent normal tissues (Figure 1B and Appendix A). Notably, *MAPT* was overexpressed in 47.3% of tumors (Figure 1C and Appendix A) and underexpressed (lower than 25% of normal levels) in 35.7% of tumors (Figure 1D and Appendix A). Whether strong variation in *MAPT* expression in tumors depends on breast cancer molecular subtypes will require further investigation with larger cohorts of patients. The two other genes, namely *MAST4* and *MTUS1*, were modestly (fold change 1.4 to 1.5) but significantly down-regulated in malignant tissues compared to normal breast tissues (Figure 1B and Appendix A). Both genes were found underexpressed in 60% of the tumors (Figure 1D and Appendix A).

Expression levels of each gene were compared among breast tumors classified according to molecular subtypes. As shown in Figure 1E and Appendix A, tumors of the TNBC subtype expressed significantly higher levels of *KIF4A*, *ASPM*, *KIF20A*, *KIF14*, *TPX2*, *KIF18B*, *KIFC1*, *AURKB*, *KIF2C*, *GTSE1*, *KIF15*, *KIF11*, *RACGAP1* and *STMN1* genes, and lower levels of *MAPT*, *MAST4* and *MTUS1* genes, compared with luminal ER-positive breast tumors.

We then explored the possibility that expression of the 17 MT-Rel genes may be co-regulated. Unsupervised hierarchical clustering of 112 breast tumors and adjacent non-tumoral breast tissue (www.tnmplot.com), based on mRNA levels of the 17 MT-Rel genes, allowed distinguishing two populations of breast tumors that differ from normal tissues (Figure 2A). In one of them (cluster 1), representing one third of breast tumors, the majority of the 17 genes were overexpressed compared to adjacent non-tumoral tissues.

Co-regulation of expression of each of the 17 genes was analyzed in more details in normal and cancer samples (Figure 2B). In normal tissues, a marked correlation (r = 0.55 to 0.94) was found between 13 up-regulated genes—*STMN1* being the exception—(Figure 2B and Appendix A), suggesting that coordinated expression of these genes may be conserved to maintain microtubule cytoskeleton homeostasis. In paired breast tumors however, correlation between the 13 up-regulated genes was weaker (r = 0.35 to 0.79, Figure 2B and Appendix A), indicating some degree of variability in the profile of MT-Rel in breast cancer, in line with the heat map shown in Figure 2A.

*MAPT*, *MAST4* and *MTUS1* formed a distinct group of genes that were not significantly correlated with any other gene (Appendix A). Notably, a weak but significant correlation between *MAST4* and *MAPT* gene expression (r = 0.37, *p* = 0.0001) was depicted in tumor samples but not in normal tissues.

We sought to determine whether MT-Rel gene expression in breast tumors may be prognostic of patient survival. Kaplan–Meyer curves of patient overall survival (OS) and relapse-free survival (RFS) from the KMplotter database (kmplot.com) revealed that each MT-Rel gene has a potential prognostic value in breast cancer. High levels of each of the 14 overexpressed genes, and low levels of *MAPT*, *MAST4* and *MTUS1*, are significantly associated with poor patient survival (Figure 3A,B, Appendix A). 

We further addressed the functional relevance of MT-Rel deregulation in breast cancer cells. To this end, a library of siRNAs targeting each of the 14 overexpressed MT-Rel genes was transfected into two breast cancer cell lines (MDA-MB-231 and MDA-MB-468) and cell viability was measured after 96 h. Results revealed that *KIF11*, *AURKB*, *TPX2* and *KIFC1* are essential genes whose depletion in both breast cancer cell lines impacts cell viability (Figure 4A,B and Appendix A). A cancer dependency map (depmap.org) further highlighted *KIF11*, *AURKB*, *RACGAP1* and *TPX2* as genes with essential effects in 47 breast cancer cell lines (Figure 4C and Appendix A). 

### 3.2. Systems Biology Analysis of Functional Interplays and Gene Networks

In a second part of our study, we investigated whether the 17 MT-Rel genes may also be functionally connected into gene networks and linked to specific biological functions and molecular mechanisms altered in cancer. To tackle this question, we undertook a Systems Biology approach centered on these 17 genes. We built the smallest network starting from the 17 MT-Rel genes, further extended to their functional partners in order to connect them within a single network, as detailed in Section 2. The extended network comprises 43 nodes (41 genes and 2 complexes) and 87 edges representing 42 functional activations, 32 inhibitions and 13 protein–protein interactions (PPI) (Figure 5A). The 17 genes and their partners are linked by edges displayed in green for activations and red for inhibitions. Formation of protein complexes is represented by gray lines. The Systems Biology approach also integrates information on differential regulation of each gene in breast cancer. Thus, among the 41 genes of the network, 26 are up-regulated (fold change > 1.5), and 8 are down-regulated (fold change < 0.66) in breast tumors compared to normal breast.

Detailed analysis of the network highlighted three major clusters (or sub-networks) in which the 17 MT-Rel genes and their partners present dense connections (Figure 5B). These clusters are related to the cellular processes associated with the following enriched GO terms: (i) spindle organization, (ii) mitotic sister chromatid segregation and (iii) cytokinesis (Figure 6). Notably, three genes (*AURKB*, *KIF4A* and *RACGAP1*) contribute to all three gene sub-networks.

Three major steps of mitosis were highlighted by the Systems Biology analysis, namely spindle assembly, chromosome segregation and cytokinesis. These cellular processes are likely controlled by fine regulation of microtubule dynamic instability.

## 4. Discussion

In this study, we have evaluated the prognostic value and functional relevance of a panel of 17 genes encoding MT-Rel proteins that were previously identified as potential predictive biomarkers of chemoresistance. Fourteen MT-Rel genes (*KIF4A*, *ASPM*, *KIF20A*, *KIF14*, *TPX2*, *KIF18B*, *KIFC1*, *AURKB*, *KIF2C*, *GTSE1*, *KIF15*, *KIF11*, *RACGAP1*, *STMN1*) were found significantly up-regulated in breast tumors compared with paired adjacent normal tissue, and were overexpressed in the aggressive TNBC subtype compared with luminal breast tumors. Notably, six of them (*KIF4A*, *ASPM*, *KIF20A*, *KIF14*, *TPX2, KIF18B*) were overexpressed by more than 10-fold. High expression of each of these genes was associated with poor clinical outcome for the patient—with both reduced overall survival and relapse-free survival—pointing to their potential value as prognostic biomarkers in breast cancer.

In line with our studies, other groups reported that *ASPM*, *KIF20A*, *TPX2*, *AURKA* and *KIF2C* are among the top 11 up-regulated hub key genes identified as potential breast cancer prognostic biomarkers [21]. *ASPM* and *AURKB* also recently appeared as key genes up-regulated in TNBC [22]. Furthermore, a meta-analysis identified *KIF20A* and *ASPM* among the top 55 overexpressed genes when comparing tumor and normal samples across the ten most frequent human cancers [14]. It is of note that the majority (8 out of 14) of up-regulated MT-Rel genes encode kinesins which are molecular motors involved in the intracellular transport of proteins and organelles along microtubules. Kinesins have been recently highlighted as prognostic biomarkers in breast cancer [23,24] and a 6-KIFs-based risk score (including four MT-Rel genes, *KIF4A*, *KIF15*, *KIF18B*, *KIF20A*) was reported to accurately predict outcomes [24].

Among the 17 MT-Rel genes studied here, 3 (*MAPT*, *MTUS1*, *MAST4*) have a different pattern of expression. They are only moderately up-regulated (MAPT) or down-regulated (*MTUS1*, *MAST4*) in breast cancer and are not co-regulated with other MT-Rel genes. Low levels of all three genes are associated with TNBC subtype, malignancy and poor prognosis for breast cancer patients, in line with previous reports for *MAPT* [25,26] and *MTUS1* [16,17], *MAST4* being much less studied.

Thus, differential regulation of all 17 MT-Rel genes in breast tumors (14 being overexpressed and 3 down-regulated) is associated with poor prognosis. 

Noticeably, 8 out of the 17 MT-Rel genes (*MAPT*, *MTUS1*, *STMN1*, *KIF2C*, *KIF18B*, *GTSE1*, *ASPM*, *KIFC1*) are involved in the regulation of microtubule dynamics and stability. Genes that are underexpressed (*MAPT*, *MTUS1*) encode microtubule stabilizers (Tau and ATIP3 proteins, respectively) [16,17,18,27] whereas genes that are overexpressed (*STMN1*, *KIF2C*, *KIF18B*) encode proteins that either destabilize [28] or depolymerize [29,30,31,32] microtubules, respectively. Other overexpressed genes (*GTSE1*, *ASPM*, *KIFC1*) indirectly control microtubule dynamics. The microtubule plus-end binding protein GTSE1 inhibits KIF2C and its overexpression increases spindle microtubule dynamics [33,34], as does the minus-end binding protein ASPM by interacting with citron kinase (CIT) [35] whereas in interphase ASPM interacts with katanin to promote severing and disassembly of dynamic microtubules [36]. Finally, the molecular motor KIFC1 (also called HSET) binds to, and disrupts, microtubule plus ends, thereby inducing catastrophe and increasing microtubule dynamic instability [37]. Globally, the emerging picture is an imbalance favoring microtubule destabilization, likely driven by the up-regulation of genes encoding microtubule destabilizing or depolymerizing proteins and down-regulation of those encoding microtubule stabilizers, with a net tendency to increase microtubule dynamics in cancer cells. This, in turn, alters the proper organization and shape of the cytoskeleton, leading to cellular abnormalities.

To further highlight cellular abnormalities driven by de-regulation of the 17 MT-Rel genes, we undertook a Systems Biology approach that extends beyond the 17 genes and takes into account differential gene expression. These studies resulted in a functional gene network that comprises 41 genes (including two protein complexes) and 87 edges contributing to major biological functions altered in breast tumors. Probing GO terms in public databases revealed that all 14 up-regulated MT-Rel genes are connected into three major functional sub-networks specifying different steps of mitosis, namely prometaphase/metaphase (control of mitotic spindle organization and integrity), anaphase (equal separation of chromosomes in each daughter cell) and cytokinesis (completion of division into two daughter cells). Defects in mitotic spindle assembly, chromosome segregation or cytokinesis likely result from altered microtubule dynamics. These defects are major drivers of aneuploidy due to improper chromosome attachment to the spindle in metaphase, chromosome lagging in anaphase and/or multinucleated cells due to cytokinesis failure. Aneuploidy and subsequent DNA damage are recognized hallmarks of cancer and are among the most important features associated with breast cancer aggressiveness.

To our surprise, despite the presence of eight kinesins and eight proteins regulating microtubule dynamics in the network, pathways involved in cell migration were not significantly highlighted in our study. Although the GO term “microtubule motor activity” (GO:0003777 with *p*-value of 9.17 × 10^−17^) reached considerable significance, the genes associated with this GO term exhibited minimal connectivity within our network. Consequently, the intracellular transport pathway would not add significant insights in terms of the Systems Biology perspective. These findings further underscore the profound impact of mitotic pathways within this gene network. 

By combining gene expression, prognostic studies, functional data and Systems Biology methods, our study points to important genes to target in breast cancer. We identify here three champions, namely *AURKB*, *TPX2* and *KIF4A*. These are highly up-regulated and/or essential genes, contributing as hubs to several functional sub-networks in deregulated breast cancer. *AURKB* encodes the mitotic kinase Aurora B that phosphorylates components at the kinetochore—where chromosomes attach microtubules—and regulates the microtubule depolymerizing activity of KIF2C [38,39]. TPX2 (Targeting Protein for Xlp2) is both a substrate and a regulator of Aurora kinase A (AURKA) that also phosphorylates a wide range of substrates in mitosis and controls the depolymerizing activity of KIF2A kinesin at the spindle pole [18,40]. *AURK* and *TPX2* have previously been identified as prognostic biomarkers of breast cancer patient survival [21]. Furthermore, both Aurora kinases and their substrates KIF2A and KIF2C are actionable proteins for which specific inhibitors have been developed in the past years [41,42,43]. Targeting Aurora kinases has been extensively explored and several clinical trials have been performed or are still ongoing, including in breast cancer, to evaluate the efficiency and the safety of Aurora kinase inhibition in cancer patients [42,44].

## 5. Conclusions

In conclusion, breast tumors with deregulated expression of MT-Rel genes are prone to cytoskeletal alterations that likely promote aneuploidy and chromosome instability. This study opens new perspectives, where targeting druggable MT-Rel proteins and their functional partners, alone or in combination with taxane-based chemotherapy, may represent an interesting therapeutical strategy in the fight against breast cancer. Together, these results may fill the gap towards the development of personalized medicine in breast cancer.

## Figures and Tables

**Figure 1 cancers-15-04870-f001:**
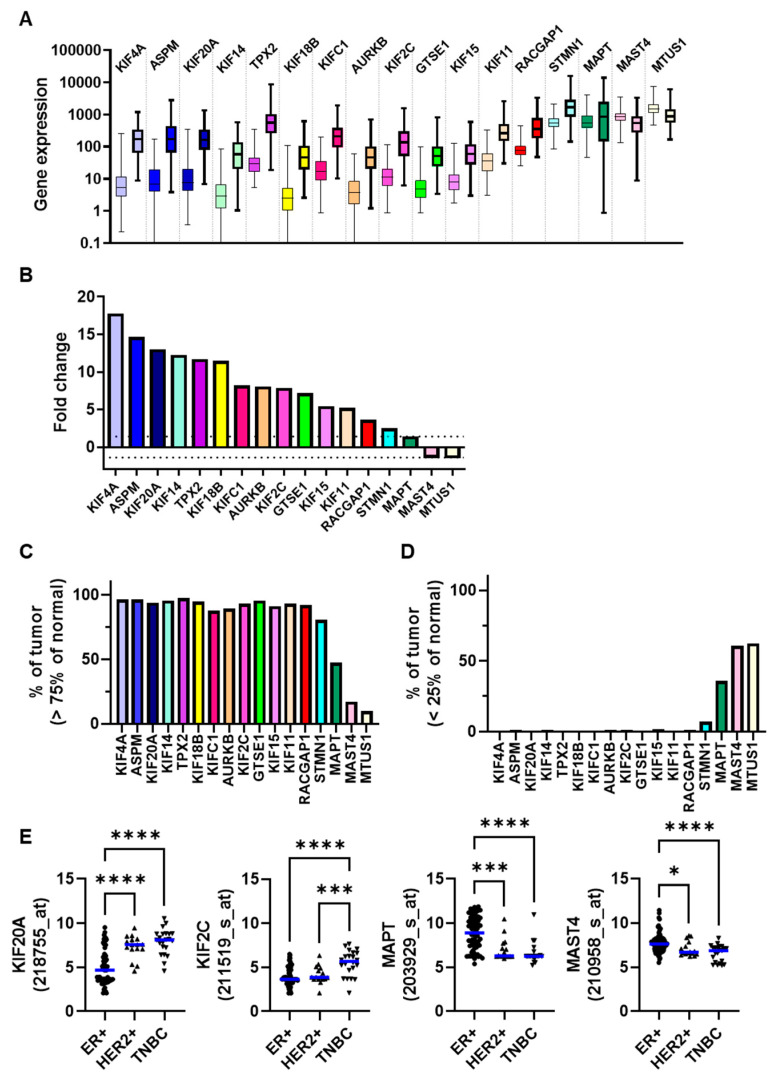
Expression of 17 MT-Rel genes in breast tumors and normal tissues. (**A**) Boxplots of mRNA expression level of the 17 MT-Rel genes in 112 breast tumors (bold boxplot) and their adjacent normal tissues (thin boxplot) from the TNMplot database (tnmplot.com). (**B**) Histograms of fold change of gene expression between tumor and normal breast tissues. Dotted line indicates a fold change value of 1.4. (**C**) Proportion of tumor samples showing higher expression of a given gene compared to normal samples using the third quartile as cutoff value. (**D**) Proportion of tumor samples showing lower expression of a given gene compared to normal samples using the first quartile as cutoff value. (**E**) Probeset intensities for each indicated gene in breast tumors from the REMAGUS02 cohort [20] classified according to their molecular subtype; ER+ (●), HER2+ (▲), TNBC (▼). A blue line indicates the median value. * *p* < 0.05; *** *p* < 0.001; **** *p* < 0.0001.

**Figure 2 cancers-15-04870-f002:**
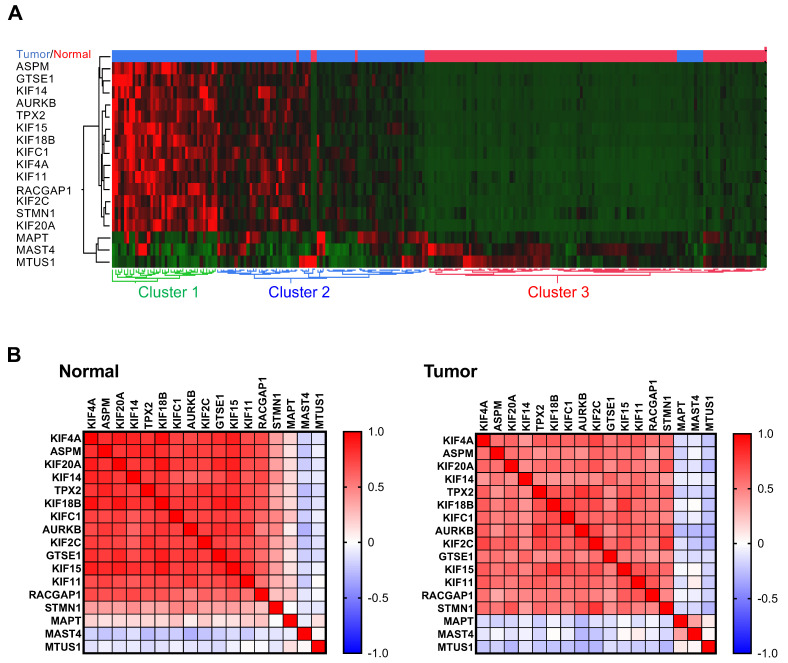
Co-regulation of expression of 17 MT-Rel genes in breast tumor and normal tissues. (**A**) Heat map hierarchical clustering of the 17 MT-Rel genes in normal (red) and tumor (blue) breast samples. Dendogram at the bottom shows the clustering of normal samples in red (cluster 3), and two clusters of tumors in blue (cluster 1) and green (cluster 2). (**B**) Heat map of Pearson correlation coefficient (r) in normal (**left**) and tumor (**right**) breast samples.

**Figure 3 cancers-15-04870-f003:**
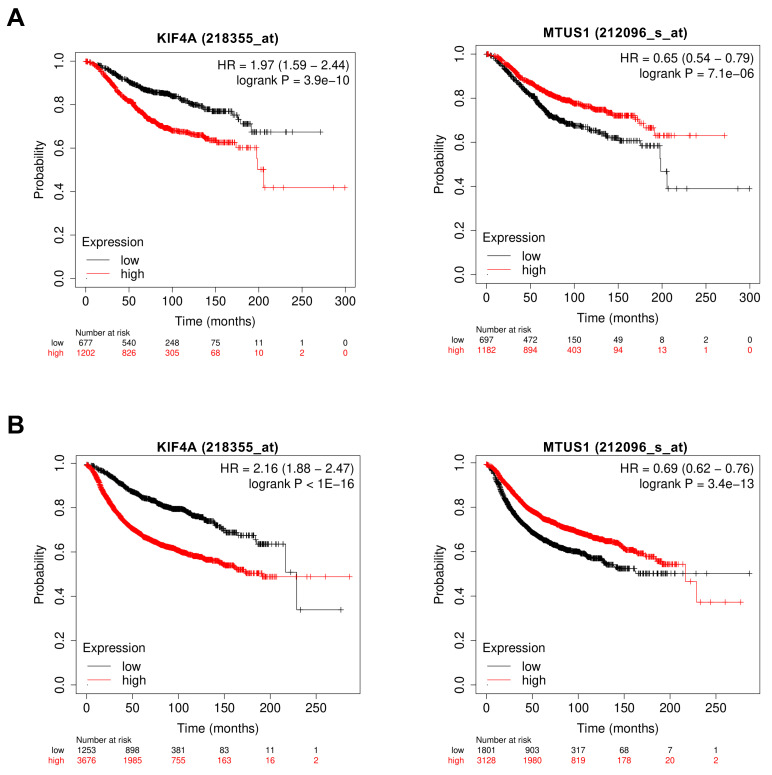
Survival curves of breast cancer patients according to MT-Rel gene expression level. (**A**) Overall survival curves of breast cancer patients according to *KIF4A* (218355_at) or *MTUS1* (212096_s_at) probeset intensities from KMplotter (kmplot.com). (**B**) Relapse-free survival curves as in (**A**).

**Figure 4 cancers-15-04870-f004:**
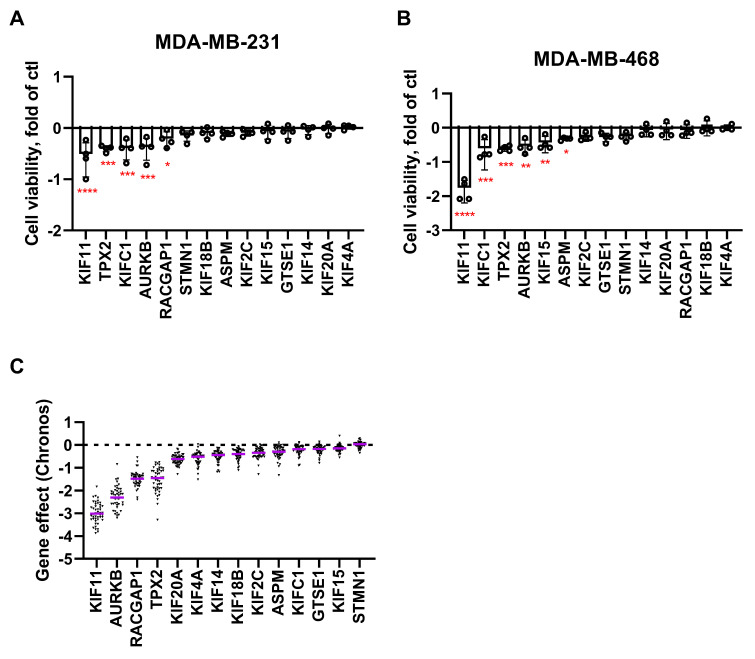
Effect of MT-Rel gene silencing on breast cancer cell viability. Cell viability was measured following silencing of each MT-Rel gene by siRNA transfection (96 h) into MDA-MB-231 (**A**) and MDA-MB-468 (**B**) breast cancer cell lines. Shown are mean values and standard deviation from four independent experiments. Significant pvalues are indicated in red; * *p* < 0.05, ** *p* < 0.01, *** *p* < 0.001, **** *p* < 0.0001. (**C**) Scattered dot plot of gene dependency score (Chronos). A lower Chronos score indicates a higher likelihood that the gene of interest is essential in a given cell line. A score of 0 indicates that a gene is non-essential. The purple lines indicate the median values.

**Figure 5 cancers-15-04870-f005:**
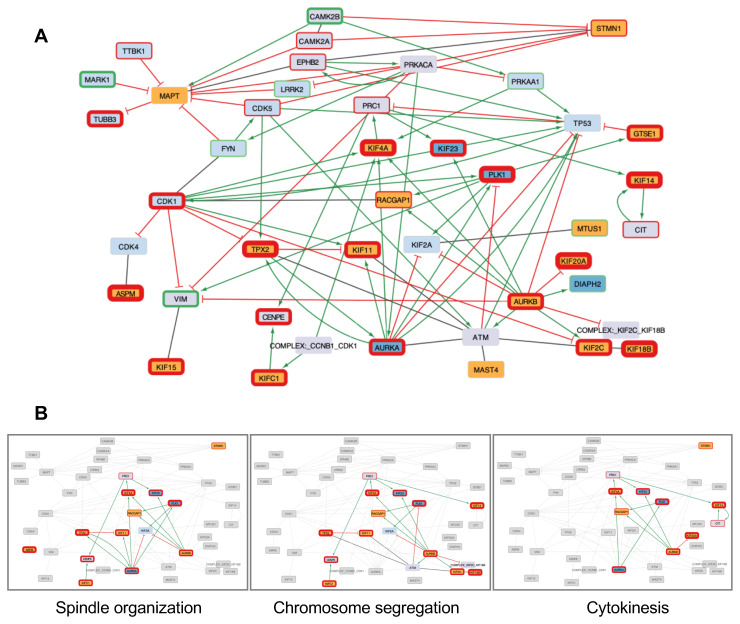
Systems Biology-derived networks connecting 17 MT-Rel genes. (**A**). The 17 MT-Rel genes (in orange) and their partners are linked by edges displayed in red for inhibitions, green for activations and black for protein–protein interactions. Node colors were assigned as follows: orange for the 17 MT-Rel genes differentially expressed in the three considered breast cancer transcriptomic datasets, blue for 17 MT-Rel genes differentially expressed in two of these datasets, light blue for genes with normal expression in breast cancer but present in the initial list of 280 MT-Rel genes and light purple for genes not present in the list of 280 MT-Rel genes. Using the tnmplot.com site, we annotated the fold change between normal and breast cancer tissues on the node: red contour if the gene is up-regulated, and green contour if the gene is down-regulated. The thicker is the border, the higher the fold change. (**B**) Sub-networks extracted from the network shown in (**A**) are associated with following enriched GO terms “spindle organization” (GO:0007051, *p* = 6 × 10^−16^) (**left**), “Mitotic sister chromatid segregation” (GO:0000070, *p* = 2 × 10^−17^) (**middle**) and “cytokinesis” GO:0000910, *p* = 3 × 10^−11^) (**right panel**).

**Figure 6 cancers-15-04870-f006:**
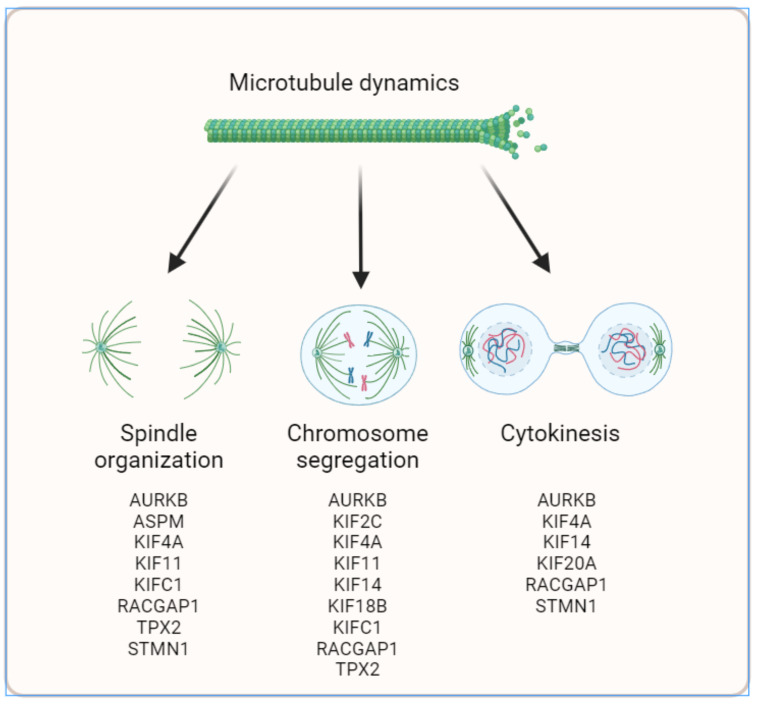
Schematic representation recapitulating the functional impact of MT-Rel genes in breast cancer.

## Data Availability

Publicly available datasets analyzed in this study can be found at the following sites: TNMplot (http://www.tnmplot.com), KMplot (http://www.kmplot.com), DepMap Public 23Q2 (https://depmap.org/portal/download/custom/), SIGNOR 3.0 database (https://signor.uniroma2.it) and Gprofiler (https://biit.cs.ut.ee/gprofiler).

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
