# Peer review of "A Network of 17 Microtubule-Related Genes Highlights Functional Deregulations in Breast Cancer"

_cancers, 2023, doi:10.3390/cancers15194870_

Round 1

Reviewer 1 Report

Microtubule dynamic instability has been long found to play an important role in the development of cancer, and antimicrotubular agents, such as paclitaxel and eribulin have been widely used in the treatment of breast cancer patients. This study evaluated the potential prognostic value of a panel of 17 MT-Rel in breast tumors and explored the functional consequence of their deregulation by using a systems biology approach. 

However, the major question for me is what gap your panel of 17 MT-Rel will fit in breast cancer clinical practice. If possible, would you provide any experimental or clinical data to convince me about the prognostic or predictive value of mitotic Aurora kinases and their substrates in the specific chemotherapy-resistant, paclitaxel for instance. Other minor concerns include:

1.     In the introduction, authors should provide brief review of the current understandings about the mechanism of resistance to chemotherapy in breast cancer, especially microtubules targeted drugs. 

2.     In line 48-49, please consider change the tone of the expressions in this sentence. There were many advancements of chemo, immuno, and target therapy for TNBC but lack of efficacy as we expected in the longer observations in general. 

3.     The main findings in reference 12 was that ATIP3 silencing in breast cancer cells potentiates the effects of paclitaxel but not the origin of MT-Rel genes mentioned in this article. In short, I may recommend authors to provide more relevant reference for the background of this study. 

Reviewer 2 Report

The article " A network of 17 microtubule-related genes highlights functional deregulations in breast cancer" study focused on a panel of 17 microtubule-related (MT-Rel) genes and their potential significance in breast cancer. The study provides a good overview of the study's objectives and findings and effectively conveys the main points of the research. It is highlighting the importance of microtubule-associated proteins and kinases in cellular regulation, particularly in cancer. Previous work of 17 MT-Rel genes in breast tumors resistant to taxane-based chemotherapy adds depth to the study and its continuity. Also, the up-regulation of 14 MT-Rel genes in breast tumors, some of which are overexpressed by a substantial margin, and the essential role of certain genes in cell survival. Additionally, the study highlights the association of these genes with poor patient survival. Nest, the results on mitotic Aurora kinases and their substrates are potential therapeutic targets for breast cancer treatment, which adds a translational aspect to the study.

Overall, the article offers valuable insights, and can be accepted as such.

English is good, can make some minor improvements.

Reviewer 3 Report

The manuscript by RODRIGUES-FERREIRA et al. showed a comprehensive analysis of 17 microtubule-related genes in breast cancer. The authors showed that expression levels of 14 microtubule-related proteins (MT-Rel) were higher in tumor samples as compared with adjacent normal tissues. The clustering analysis showed the co-regulation of these MT-Rel genes. Overall survival and relapse-free survival were examined among MT-Rel genes, and they were associated with poor prognosis. Loss-of-function experiment with each siRNA revealed that KIF11, AURKB, TPX2 214 and KIFC1 are important for cell survival in breast cancer cell lines. Finally, gene networks with 17 MT-Rel genes were investigated based on systems biology approach, and three major clusters such as spindle organization, mitotic sister chromatid segregation, and cytokinesis were identified.

1.       In Figure 1, expression levels of MT-Rel genes were shown in breast cancer samples, both tumor and normal. It would be more informative to show these data based on breast cancer subtypes such as luminal, HER2 basal, and TNBC, because subtype classification is closely related to clinical applications in breast cancer.

2.       In Figure 4, cell viability was examined after knocking down of each MT-Rel gene. However, knockdown efficiency of each MT-Rel gene was not shown. The data of knockdown % should be shown. If possible, more than two different siRNA sequences should be used to avoid off-target effects.

3.       The authors mentioned “KIF11, AURKB, TPX2 214 and KIFC1 are essential genes whose depletion in both breast cancer cell lines leads to cell 215 death”. No direct evidence of cell death was shown in the manuscript. Have the authors examined the rate of apoptotic cell death?

4.       In Figure 5, systems biology-derived networks connecting 17 MT-Rel genes showed 3 major networks such as spindle organization, mitotic sister chromatid segregation, and cytokinesis. Because these networks were made with microtubule-related proteins, it is not unexpected data showing microtubule-related networks. Thus, it is unclear what is critical points of this analysis. Also, it is not related to breast cancer biology. 

Round 2

Reviewer 3 Report

The authors properly revised the manuscript.